# Prognostic Role and Clinical Significance of Tumor-Infiltrating Lymphocyte (TIL) and Programmed Death Ligand 1 (PD-L1) Expression in Triple-Negative Breast Cancer (TNBC): A Systematic Review and Meta-Analysis Study

**DOI:** 10.3390/diagnostics10090704

**Published:** 2020-09-17

**Authors:** Parisa Lotfinejad, Mohammad Asghari Jafarabadi, Mahdi Abdoli Shadbad, Tohid Kazemi, Fariba Pashazadeh, Siamak Sandoghchian Shotorbani, Farhad Jadidi Niaragh, Amir Baghbanzadeh, Nafiseh Vahed, Nicola Silvestris, Behzad Baradaran

**Affiliations:** 1Research Center for Evidence-Based Medicine, Tabriz University of Medical Sciences, Tabriz 5166614766, Iran; p.lotfinezhad@gmail.com (P.L.); fariba.pashazadeh@gmail.com (F.P.); 2Immunology Research Center, Tabriz University of Medical Sciences, Tabriz 5165665811, Iran; abdoli.med99@gmail.com (M.A.S.); amirbaghbanzadeh@gmail.com (A.B.); 3Department of Immunology, Tabriz University of Medical Sciences, Tabriz 5166614766, Iran; kazemit@tbzmed.ac.ir (T.K.); siamak1331@gmail.com (S.S.S.); farhad_jadidi1986@yahoo.com (F.J.N.); 4Student Research Committee, Tabriz University of Medical Sciences, Tabriz 5165665811, Iran; 5Department of Statistics and Epidemiology, Faculty of Health, Tabriz University of Medical Sciences, Tabriz 5165665931, Iran; m.asghari862@gmail.com; 6Emergency Medicine Research Team, Tabriz University of Medical Sciences, Tabriz 5166614766, Iran; vahedn66@gmail.com; 7Medical Oncology Unit, IRCCS Istituto Tumori “Giovanni Paolo II” of Bari, 70124 Bari, Italy; 8Department of Biomedical Sciences and Human Oncology (DIMO), University of Bari, 70124 Bari, Italy; 9Pharmaceutical Analysis Research Center, Tabriz University of Medical Sciences, Tabriz 5165665811, Iran

**Keywords:** programmed cell death-ligand 1, tumor-infiltrating lymphocytes, triple-negative breast cancer, prognosis, meta-analysis

## Abstract

This meta-analysis aimed to evaluate the prognostic value of tumor-infiltrating lymphocytes (TILs) and programmed death-ligand 1 (PD-L1), their associations with the clinicopathological characteristics, and the association between their levels in patients with triple-negative breast cancer (TNBC). PubMed, EMBASE, Scopus, ProQuest, Web of Science, and Cochrane Library databases were searched to obtain the relevant papers. Seven studies with 1152 patients were included in this study. Like the level of TILs, there were no significant associations between PD-L1 expression and tumor size, tumor stage, lymph node metastasis, histological grade, and Ki67 (All *p*-values ≥ 0.05). Furthermore, there was no significant association between PD-L1 expression with overall survival (OS) and disease-free survival (DFS). In assessment of TILs and survival relationship, the results showed that a high level of TILs was associated with long-term OS (hazard ratios (HR) = 0.48, 95% CI: 0.30 to 0.77, *p*-value < 0.001) and DFS (HR = 0.53, 95% CI: 0.35 to 0.78, *p*-value < 0.001). The results displayed that tumoral PD-L1 expression was strongly associated with high levels of TILs in TNBC patients (OR = 8.34, 95% CI: 2.68 to 25.95, *p*-value < 0.001). In conclusion, the study has shown the prognostic value of TILs and a strong association between tumoral PD-L1 overexpression with TILs in TNBC patients.

## 1. Introduction

Breast cancer is the leading cause of mortality among females worldwide [1]. Triple-negative breast cancer (TNBC) is responsible for 10% to 20% of total breast cancer cases. TNBC cells do not express the conventional receptors, i.e., estrogen receptor, human epidermal growth factor-2, and progesterone receptor [2]. Therefore, target therapies have failed to bring desired outcomes in patients with TNBC. Indeed, TNBC patients have poorer outcomes than patients with other breast cancer types [3,4].

Tumor-infiltrating lymphocytes (TILs) are responsible for developing anti-tumoral immune responses. TILs can recognize the tumoral antigens and eliminate the tumoral cells [5]. However, tumor cells can induce an immunosuppressive tumor microenvironment and evade the anti-tumoral immune responses. The aim of immunotherapy is restoring the anti-tumoral immune responses to reject tumoral cells [6,7]. Immune checkpoints have been implicated in the induction of immunosuppressive tumor microenvironments [7]. Programmed cell death protein 1 (PD-1) can overexpress on the TILs and pave the road for suppressing anti-tumoral immune responses [8]. According to preclinical studies, the programmed death-ligand 1 (PD-1/PD-L1) axis substantially “exhausts” TILs, increases the recruitment of inhibitory cells, and finally suppresses anti-tumoral immune responses [9]. However, the PD-L1/PD-1 axis is not the only recognized immune checkpoint axis in the dynamic tumor microenvironment of TNBC; multiple immune checkpoints and immune cells can direct the anti-tumoral immune responses of TILs in affected patients [10].

TNBC cells can overexpress PD-L1 on the cell surface and induce tolerance against tumoral antigens [11]. Moreover, a substantial increase in the level of TILs has been noted in patients with TNBC [12]. Although multiple studies have investigated the association between tumoral PD-L1 expression and TILs with the prognosis of patients with TNBC, there is a controversy about the prognostic nature of PD-L1 expression and TILs and their clinicopathological relevance in patients with TNBC [13,14,15,16,17,18,19,20].

Therefore, there is an urgent need to determine the prognostic values of tumoral PD-L1, TILs, and their associations with the clinicopathological features in TNBC patients. To the best of our knowledge, this meta-analysis, for the first time, has aimed to determine the prognostic values of tumoral PD-L1, TILs, and their associations with clinicopathological features in TNBC patients.

## 2. Materials and Methods

This study was performed under the Preferred Reporting Items for Systematic Reviews and Meta-Analyses (PRISMA) statement [21]. In the current study, a meta-analysis was performed on previous studies, so there is no necessity to obtain the patient’s consent and ethical approval.

### 2.1. Search Strategy

The PubMed, EMBASE, Scopus, ProQuest, Web of Science, and Cochrane Library databases were searched systematically for relevant papers before 30 June 2020. The systematic search has used the following keywords: (“PD-L1” OR “B7-H1” OR “CD274 “ OR “programmed cell death 1 ligand 1” OR “B7H1” OR “immune checkpoint”) AND (“breast cancer” OR “breast tumor” OR “cancer of the breast” OR “human mammary neoplasm” OR “human mammary carcinomas” OR “TNBC”) AND (“tumor infiltrating lymphocytes” OR “TIL“ OR “T-cells” OR “T-lymphocytes”). Furthermore, the references of previous related review studies have been screened to identify the potentially eligible papers.

### 2.2. Eligibility Criteria

Records with the following eligibility criteria were included in this analysis: (1) original papers in English, (2) studies with the objective of assessment TILs and tumor cells and PD-L1 expression in TNBC patients, (3) studies in which PD-L1 expression and TILs were determined by hematoxylin and eosin (H&E) or immunohistochemistry (IHC) techniques and PD-L1 expression was determined by IHC techniques, (4) records that explain TILs and PD-L1 association with the prognosis of patients with TNBC, (5) papers demonstrating the correlation of TILs and PD-L1 expression with clinicopathological characteristics, and (6) studies that provide the data for hazard ratios (HRs) and 95% CI extraction for overall survival (OS) and disease-free survival (DFS) analysis. Records were excluded from this study according to the following criteria: studies that failed to meet the aforementioned inclusion criteria, non-human experiments, studies in which the TILs and PD-L1 were evaluated after treatment, duplicate studies, review papers, case report studies, and conference abstracts.

### 2.3. Data Extraction

After discarding the duplicated papers, two authors independently reviewed the titles and abstracts of all identified records. All disagreements were resolved by consensus and consultation with the third reviewer. The following data were extracted from included studies: the first author, publication year, country of study, patients’ number, TILs and PD-L1 detection method, cut-off values of the high/positive rates for TILs and PD-L1 expression, and follow-up time. Survival data, including HR and 95% CI for OS and DFS, were extracted from relevant papers for further analysis.

### 2.4. Quality Assessment of Studies

The quality of included studies was assessed based on Hayden et al. guidelines for evaluating quality in prognostic studies [22].

### 2.5. Statistical Analysis

All analyses were conducted by STATA16 (StataCorp, College Station, TX, USA). The study statistician performed data extraction for primary outcomes. Random effect Meta-analyses were performed using the Restricted Maximum Likelihood Method (REML) [23]. The random effect model was used because there may be other unknown, unregistered/unpublished studies that we could not have had access to. The between-study heterogeneity was evaluated using the Cochran Q test, Tau-squared, H-Squared, and I-Squared statistics. Significance results of the test and values higher than 75% for I-squared were considered as substantial heterogeneity, while a value of H-Squared = 1 indicated perfect homogeneity between studies [24]. The common effect sizes were calculated as the OR to evaluate the relationship between TILs and expression of PD-L1 with the clinicopathological characteristics. HR was used to evaluate the relationship between TILs and PD-L1 expression with the OS and DFS outcomes. The 95% CIs for OR and HR were estimated. The funnel plots were drawn to assess the publication bias.

Furthermore, Egger’s [25] and Begg’s [26] were conducted to assess the bias. The meta-regression analysis is usually required to determine the source of heterogeneity. However, it was not performed in the current study due to the small sample size. Moreover, to assess the effect of individual studies on effect size (ES), sensitivity analyses were conducted, which showed no outlier study.

## 3. Results

### 3.1. Search Results

The process of literature selection is summarized in (Figure 1). The comprehensive systematic search identified 2806 potentially relevant records. After initial screening, 1540 duplicated studies were excluded. After reviewing the eligibility criteria, 1175 studies were excluded based on reviewing their titles and abstracts. The full texts of the remaining 91 records were assessed, and 84 articles were excluded for the following reasons: non-human studies, studies evaluating PD-L1 expression on the surface of other cells, no-TILs investigation, and insufficient documented data for analysis. Ultimately, a total of 7 studies remained [13,14,15,16,17,18,27]. The total sample size was 1152 participants. Most of the studies did not provide data regarding the experience of patients on being immune checkpoint inhibitors. Table 1 aims to summarize the main features of the included studies.

### 3.2. Features of Included Studies

The main features of the included studies are provided in (Table 1). In all seven studies, PD-L1 expression was assessed by IHC techniques, and IHC or H&E methods were performed for TIL evaluation. Studies had a different scoring system for both PD-L1 and TILs measurements. The median follow-up time of included studies was from 55 to 127.3 months. DFS and OS were reported as surrogate endpoints in 5 studies, and 1 study used just OS as an endpoint factor. Additionally, one study did not report OS and DFS data as HR and 95% CI.

### 3.3. Expression of PD-L1 and Clinicopathological Features

The current meta-analysis has evaluated the relationship between PD-L1 expression on tumor cells and clinicopathological features. The results have demonstrated that there are no significant associations between PD-L1 expression and tumor size (OR = 0.98, 95% CI: 0.70 to 1.36, *p*-value = 0.89) [14,15,17,18,27], tumor stage (OR = 0.70, 95% CI: 0.29 to 1.67, *p*-value = 0.42) [14,15,17,27], lymph node metastasis (OR = 1.06, 95% CI: 0.75 to 1.49, *p*-value = 0.74) [14,15,17,18,27], histological grade (OR = 1.49, 95% CI: 0.68 to 3.29, *p*-value = 0.32) [14,15,18,27], and Ki67 (OR = 1.16, 95% CI: 1.00 to 1.35, *p*-value = 0.05) [15,27] (All *p*-values ≥ 0.05). Figure 2 shows the forest plot of individual effect sizes within each study.

### 3.4. TILs and Clinicopathological Features

Additionally, the meta-analysis was performed to investigate the association between TILs and clinicopathological characteristics. The results have elucidated that there is no significant association between increased TILs and tumor size (OR = 1.09, 95% CI: 0.75 to 1.59, *p*-value = 0.63) [14,17,27], tumor stage (OR = 1.96, 95% CI: 0.98 to 3.91, *p*-value = 0.06) [14,17,27], and lymph node metastasis (OR = 1.18, 95% CI: 0.66 to 2.13, *p*-value = 0.57) [14,17,27]. (All *p*-values > 0.05). Figure 3 shows the forest plot of individual effect sizes within each study.

### 3.5. Relationship between PD-L1 Expression and Survival

The meta-analysis of five studies has shown that there is no significant association between PD-L1 overexpression and OS (HR = 0.56, 95% CI: 0.32 to 1.01, *p*-value = 0.056) [14,15,16,17,18], and there is no significant association between PD-L1 expression and DFS (HR = 0.65, 95% CI: 0.37 to 1.16, *p*-value = 0.15) [14,15,16,17,18]. (All *p*-values > 0.05). Figure 4 shows the forest plot of individual effect sizes within each study.

### 3.6. Relationship between TILs and Survival

Moreover, the current study has demonstrated that TILs are indicators of prolonged OS (HR = 0.48, 95% CI: 0.30 to 0.77, *p*-value < 0.001) [13,14,15,16,17,18] and DFS (HR = 0.53, 95% CI: 0.35 to 0.78, *p*-value < 0.001) [14,15,16,17,18]. Figure 5 shows the forest plot of individual effect sizes within each study.

### 3.7. Relationship between PD-L1 Expression and TILs

The meta-analysis was conducted to investigate the relationship between TILs and PD-L1 expression on tumor cells. The results have reported that PD-L1 overexpression is strongly related to TILs in TNBC patients (OR = 8.34, 95% CI: 2.68 to 25.95, *p*-value < 0.001) (Figure 6). 

### 3.8. Publication Bias

The Egger’s and Begg’s tests were performed to evaluate the probability of the publication bias. These tests have shown no significant small-study effect and publication bias (All *p*-values > 0.05). The results from funnel plot analysis (Figure 7, Appendix A) demonstrate that asymmetry is not present, and there is no publication bias affecting the ORs and HRs.

### 3.9. Bias Assessment in Included Studies

Furthermore, the quality of the included studies was assessed under six items of Hayden et al. guidelines. All mentioned items of the guidelines were fulfilled by all studies, except for “blind measurement” in the study by Okabe et al., and “clarity of cut-off positivity” in the study by Yeong et al. (Figure 8).

## 4. Discussion

Multiple studies have investigated the association between tumoral PD-L1 expression and TILs with the prognosis of patients with TNBC. However, there is a controversy about the prognostic nature of PD-L1 expression and TILs in patients with TNBC [13,14,15,16,17,18]. As far as we know, it is the first meta-analysis that has intended to resolve this controversy.

Our analysis has displayed that there has been no association between tumoral PD-L1 expression and a better prognosis in patients with TNBC. Although a meta-analysis has shown that tumoral PD-L1 expression is associated with a poor prognosis in patients with breast cancer (HR, 1.63; 95% CI, 1.07–2.46; *p* = 0.02), this study has failed to demonstrate any significant HR between tumoral PD-L1 expression and a better prognosis in TNBC patients. (*p*-value ≥ 0.05) [28]. Since the tumor microenvironment of TNBC is regulated with multiple immune checkpoints and immune cells, the measurement of only PD-L1 expression might not be a solid prognostic value [10]. Indeed, PD-L1, on its own, may not serve as a reliable prognostic biomarker. However, it has a notable prediction value when reported with other indicators, e.g., TILs [29].

Our study has demonstrated that a high level of TILs is associated with increased long-term OS and DFS in TNBC patients. Consistent with our study, another meta-analysis has highlighted that the high level of TILs is associated with a better prognosis in TNBC patients [19]. Moreover, Gao et al. has reported that increased TILs conferred an improved survival rate. In patients with TNBC, CD4^+^, CD8^+^, and FOXP3^+^ lymphocytes have been indicative of a favorable prognosis [30].

We have also analyzed the potential association between PD-L1 expression and TILs with clinicopathological features in TNBC patients. According to our pooled analysis, there has been no significant association between PD-L1 expression and clinicopathological features (all *p*-values were higher than 0.05 (*p*-value ≥ 0.05)). Furthermore, there has been no significant association between TILs and clinicopathological features. Therefore, increased TILs have not significantly pertained to increased tumor size, higher tumor grade, and positive lymph node metastasis.

Moreover, in the current study, we have shown a strong association between PD-L1 expression and TILs. Consistent with this, Bae et al. have discovered that high expression of PD-L1 was significantly related to high TILs levels [31]. Increased TILs levels eliminate the tumoral cells and can predict a better response rate to immunotherapy. Increased PD-L1 expression and an up-regulated level of TILs have provided ample opportunity for the immunotherapy to eliminate the tumoral cells [17]. Consistent with this, immune checkpoint therapy has shown promising results in patients with TNBC [32].

This current meta-analysis has several strengths. First, as far as we can tell, it is the first meta-analysis that has comprehensively elucidated the impact of TILs and tumoral PD-L1 expression on the prognosis and clinicopathological features of TNBC patients. Second, this study has highlighted a strong association between tumoral PD-L1 expression and TILs. Third, highlighting this strong association with cancers like TNBC, which do not have well-established targets, is even more valuable. However, this meta-analysis has several limitations. First, only the papers in English were included, while other published records were omitted. Second, a small number of studies were included in the analysis, i.e., studies evaluated TILs with different methods (IHC and H&E), the studies utilized variable cut-off values for assessing PD-L1 expression, and investigations used different scoring systems for evaluating TILs. These mentioned limitations probably lead to heterogeneity between the included studies.

## 5. Conclusions

This study is intended to highlight the prognostic values of TILs and tumoral PD-L1 expression and their associations with clinicopathological features in patients with TNBC. According to this study, increased levels of TILs have been associated with a better prognosis in TNBC patients. From a clinical perspective, there has been a strong association between the tumoral PD-L1 expression and TILs. This strong association can serve as a pivotal target for cancer immunotherapy in patients with TNBC.

## Figures and Tables

**Figure 1 diagnostics-10-00704-f001:**
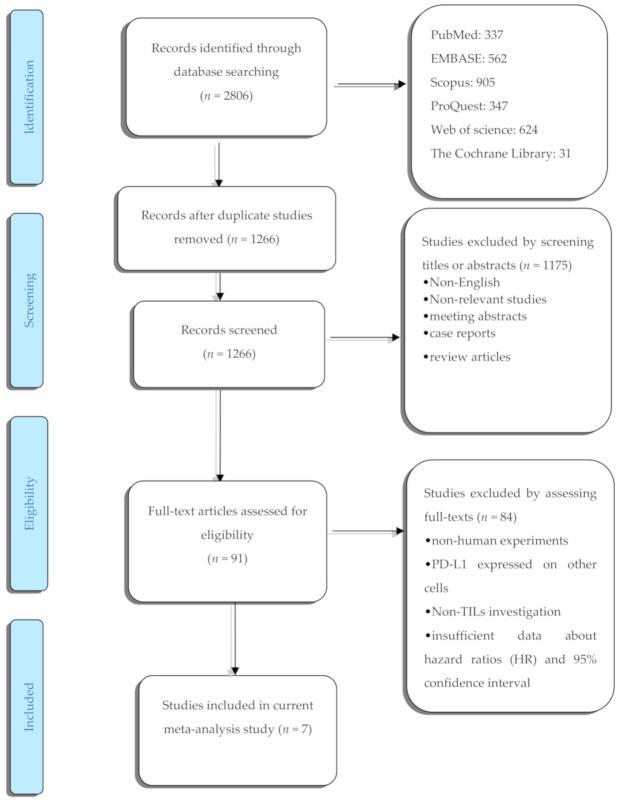
The flow diagram of study.

**Figure 2 diagnostics-10-00704-f002:**
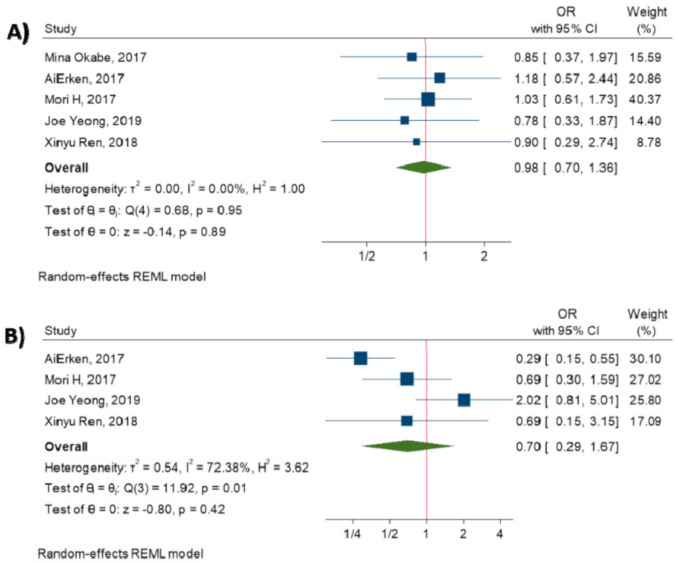
Forest plot of the individual odds ratio for the relation between PD-L1 expression and clinicopathological characteristics. (**A**) tumor size, (**B**) tumor stage, (**C**) lymph node metastasis, (**D**) histological grade, (**E**) Ki67.

**Figure 3 diagnostics-10-00704-f003:**
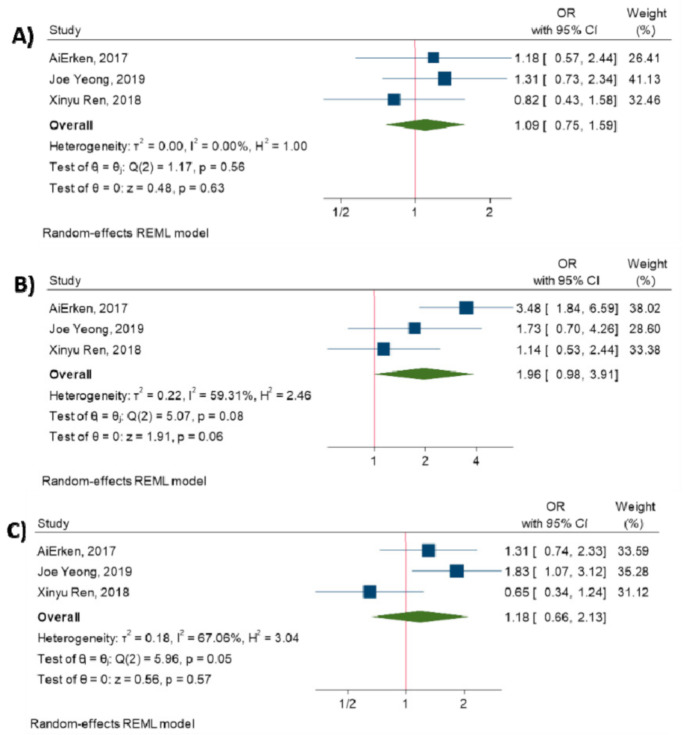
Forest plot of the individual odds ratio for the relation between TILs and clinicopathological characteristics. (**A**) tumor size, (**B**) tumor stage, (**C**) lymph node metastasis.

**Figure 4 diagnostics-10-00704-f004:**
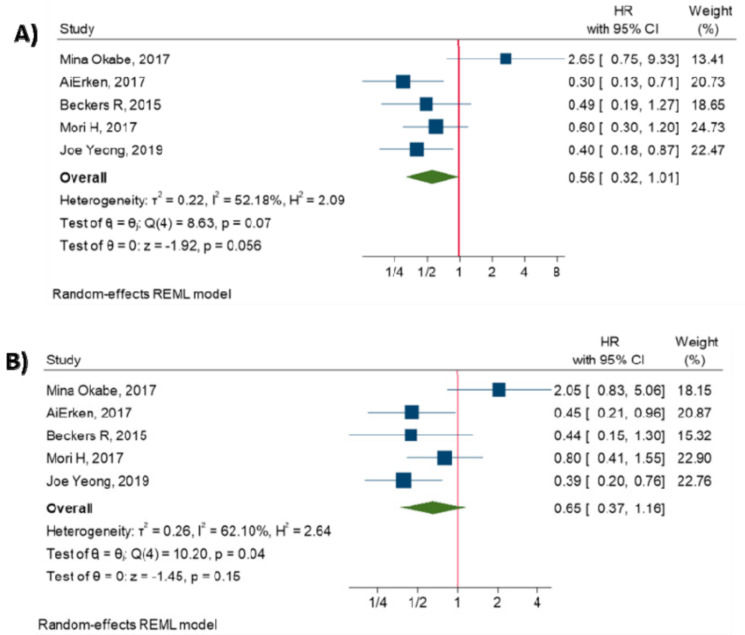
Forest plot of individual relative risks for PD-L1 expression-related survival. (**A**) OS. (**B**) DFS.

**Figure 5 diagnostics-10-00704-f005:**
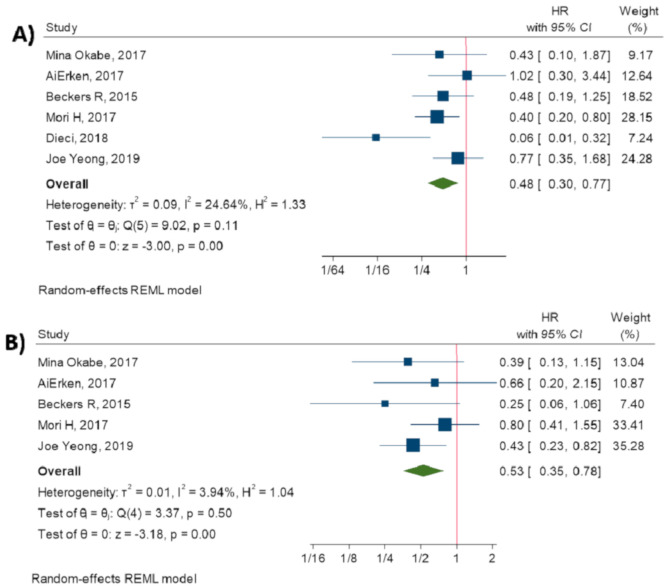
Forest plot of individual relative risks for TILs related survival. (**A**) OS. (**B**) DFS.

**Figure 6 diagnostics-10-00704-f006:**
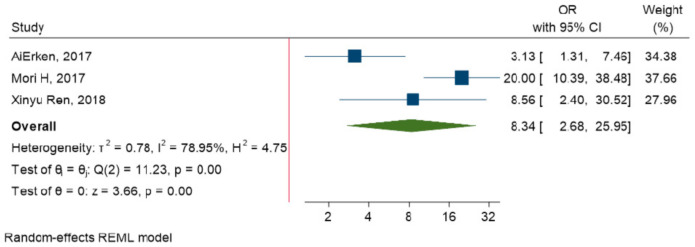
Forest plot of the individual odds ratio for the relation between TILs and PD-L1 expression.

**Figure 7 diagnostics-10-00704-f007:**
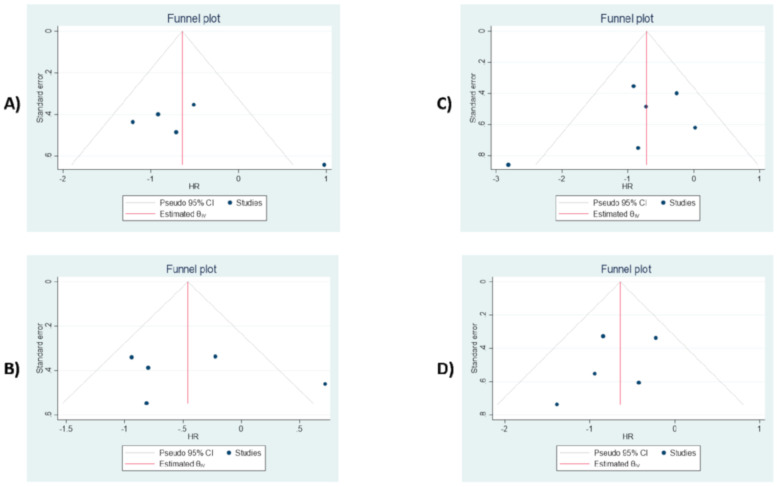
Funnel plot assessment of possible publication bias. (**A**) PD-L1 expression for OS. (**B**) PD-L1 expression for DFS. (**C**) TILs for OS. (**D**) TILs for DFS.

**Figure 8 diagnostics-10-00704-f008:**
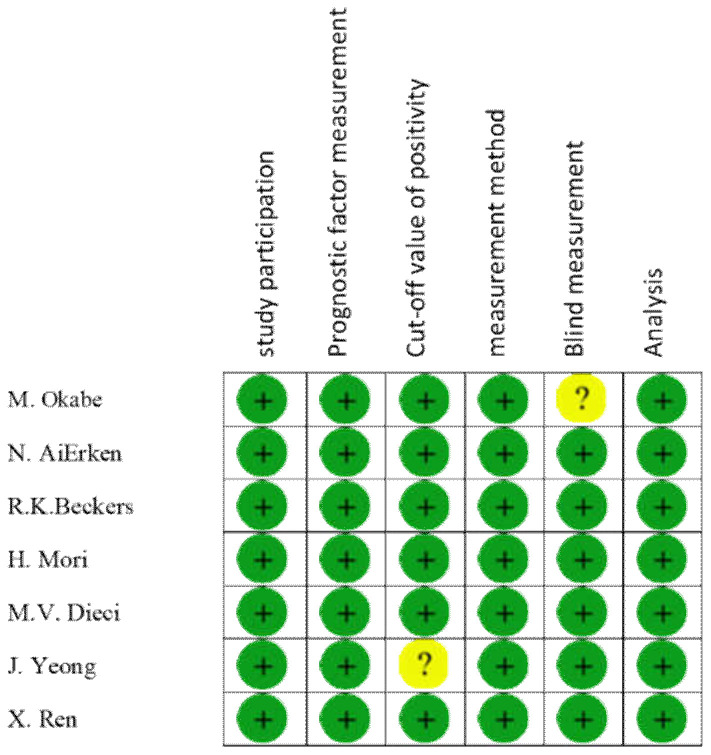
Risk of bias assessment in 7 studies.

**Table 1 diagnostics-10-00704-t001:** Main features of the included studies in the current meta-analysis.

	Author	Year	Country	Number of Patients	Cut-Off for PD-L1 (Positive/High Expression)	Cut-Off for TIL (Positive/High Expression)	PD-L1 Evaluation Method	TILs Evaluation Method	Follow-Up Time	Endpoint	Number of Patients with Experience of ICIs
1	M. Okabe [18]	2017	Japan	21	H-score > 3	≥50%	IHC	IHC	127.3	OS/DFS	0
2	N. AiErken [17]	2017	China	215	positive (>1)	(41% to 100%)	IHC	IHC	67.7 (7–159)	OS/DFS	N/a
3	R. K. Beckers [16]	2015	Australia	161	H-score >100	(score 3) > 60%	IHC	H&E	55 (0–213)	OS/DFS	N/a
4	H. Mori [15]	2017	Japan	248	≥50%	≥50%	IHC	H&E	68 (2–150)	OS/DFS	N/a
5	M. V. Dieci [13]	2018	Italy	43	≥5%	≥10%	IHC	H&E	unclear	OS	N/a
6	J. Yeong [14]	2019	Singapore	269	unclear	unclear	IHC	IHC	97 (1–213)	OS/DFS	N/a
7	X. Ren [27]	2018	China	195	>25%	(score 3) > 60%	IHC	H&E	unclear	OS/DFS	N/a

Abbreviations: PD-L1: programmed death-ligand 1, TIL: tumor-infiltrating lymphocytes, ICIs: immune checkpoint inhibitors, N/a: not available, IHC: immunohistochemistry, H&E: hematoxylin and eosin.

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
