# Peer review of "Prognostic Role and Clinical Significance of Tumor-Infiltrating Lymphocyte (TIL) and Programmed Death Ligand 1 (PD-L1) Expression in Triple-Negative Breast Cancer (TNBC): A Systematic Review and Meta-Analysis Study"

_diagnostics, 2020, doi:10.3390/diagnostics10090704_

Round 1
Reviewer 1 Report
This review article entitled “Prognostic Role and Clinical Significance of Tumor-Infiltrating Lymphocyte (TIL) and Programmed Death Ligand 1 (PD-L1) expression in Triple-Negative Breast Cancer (TNBC): A systematic review and meta-analysis study” by Parisa Lotfinejad et al. is a comprehensive review and meta-analysis about prognostic and clinical significance of TIL and PD-L1 expression in TNBC. The authors found that the expression of PD-L1 with highly TILs infiltration was related to better prognosis in TNBC patients and increased expression of PD-L1 was strongly related to high levels of TILs. The descriptions on this paper are correctly based on the previous papers. The beautiful figures and well-organized tables on the current form will help readers understand the contents easily and precisely. This paper will be of interest to readers of Diagnostics. I have only a few concerns mentioned below.
Comments
1. The article describes the correlation between PD-L1 expression and prognosis. It is necessary to show whether immunotherapy including immune checkpoint inhibitors was applied to patients in the papers adopted in this meta-analysis.
2. English used in the current form contains a significant number of grammatical errors and typos. The authors should revise it in a fundamental way after consultation with a native English speaker.
Author Response
Reviewer #1:
This review article entitled "Prognostic Role and Clinical Significance of Tumor-Infiltrating Lymphocyte (TIL) and Programmed Death Ligand 1 (PD-L1) expression in Triple-Negative Breast Cancer (TNBC): A systematic review and meta-analysis study" by Parisa Lotfinejad et al. is a comprehensive review and meta-analysis about prognostic and clinical significance of TIL and PD-L1 expression in TNBC. The authors found that the expression of PD-L1 with highly TILs infiltration was related to better prognosis in TNBC patients and increased expression of PD-L1 was strongly related to high levels of TILs. The descriptions on this paper are correctly based on the previous papers. The beautiful figures and well-organized tables on the current form will help readers understand the contents easily and precisely. This paper will be of interest to readers of Diagnostics. I have only a few concerns mentioned below.
- The article describes the correlation between PD-L1 expression and prognosis. It is necessary to show whether immunotherapy including immune checkpoint inhibitors was applied to patients in the papers adopted in this meta-analysis.
We are appreciative of the valuable comment. We added the requested information both on the text and table and revised the table to help the readers to understand the content precisely.
Response to the comment:
Most of the studies did not provide data regarding the experience of patients on being immune checkpoint inhibitors.
- English used in the current form contains a significant number of grammatical errors and typos. The authors should revise it in a fundamental way after consultation with a native English speaker.
We appreciate the constructive comment of the reviewer. A native English speaker has revised the manuscript and corrected those grammatical errors.
Reviewer 2 Report
This study by Lotfinejad et al. aims to evaluate the prognostic relationship between Tumor-Infiltrating Lymphocytes (TILs) and PD-L1 expression in survival of triple-negative breast cancer (TNBC) patients. The authors sought to carry out a meta-analysis of data of research studies published in PubMed, EMBASE, Scopus, ProQuest, Web of Science, and Cochrane Library databases. However, their findings are not innovative. The Introduction is currently too short to efficiently and adequately provide the readers with the background. The Discussion fails also to discuss the importance of the authors’ findings and important references from the literature have also been neglected. Finally, the Reviewer believes that this meta-analysis does not significantly contribute to the existing knowledge on the field, and concludes that this manuscript does not merit to be published in Diagnostics.
Author Response
Reviewer #2:
This study by Lotfinejad et al. aims to evaluate the prognostic relationship between Tumor-Infiltrating Lymphocytes (TILs) and PD-L1 expression in survival of triple-negative breast cancer (TNBC) patients. The authors sought to carry out a meta-analysis of data of research studies published in PubMed, EMBASE, Scopus, ProQuest, Web of Science, and Cochrane Library databases. However, their findings are not innovative. The Introduction is currently too short to efficiently and adequately provide the readers with the background. The Discussion fails also to discuss the importance of the authors' findings and important references from the literature have also been neglected. Finally, the reviewer believes that this meta-analysis does not significantly contribute to the existing knowledge on the field, and concludes that this manuscript does not merit to be published in Diagnostics.
We want to extend our appreciation for this comment. We performed the current meta-analysis study about combined PD‐L1 and TIL effect as comprehensive immuno‐oncological integral biomarkers on the survival values of TNBC patients. Since each of them alone has been proven to be an influential determinant in the outcome of TNBC patients, we have extracted data from research papers in which both of TILs and PD-L1 expression were evaluated. Indeed, our meta-analysis has highlighted the prognostic value of TILs and a strong association between TILs and tumoral PD-L1 expression. Hence, we have highlighted the originality and superiority of the current meta-analysis over others. Furthermore, we have revised the abovementioned sections, according to the reviewer's comment.
Response to the comment:
Introduction
Breast cancer is the leading cause of mortality among females worldwide (1). TNBC is responsible for 10% to 20% of total breast cancer cases. TNBC cells do not express the conventional receptors, i.e., estrogen receptor, human epidermal growth factor-2, and progesterone receptor (2). Therefore, target therapies have failed to bring desired outcomes in patients with TNBC. Indeed, TNBC patients have poor outcomes than patients with other breast cancer types (3, 4).
TILs are responsible for developing anti-tumoral immune responses. TILs can recognize the tumoral antigens and eliminate the tumoral cells (5). However, tumor cells can induce an immunosuppressive tumor micro-environment and evade the anti-tumoral immune responses. The aim of immunotherapy is restoring the anti-tumoral immune responses to reject tumoral cells (6, 7). Immune checkpoints have been implicated in the induction of immunosuppressive tumor microenvironment (7). PD-1 can overexpress on the TILs and pave the road for suppressing anti-tumoral immune responses (8). According to preclinical studies, the PD-1/PD-L1 axis substantially "exhausts" TILs, increases the recruitment of inhibitory cells, and finally suppresses anti-tumoral immune responses (9). However, the PD-L1/PD-1 axis is not the only recognized immune checkpoint axis in the dynamic tumor microenvironment of TNBC; multiple immune checkpoints and immune cells can direct the anti-tumoral immune responses of TILs in affected patients (10).
TNBC cells can overexpress PD-L1 on the cell surface and induce tolerance against tumoral antigens (11). Moreover, a substantial increase in the level of TILs has been noted in patients with TNBC (12). Although multiple studies have investigated the association between tumoral PD-L1 expression and TILs with the prognosis of patients with TNBC, there is a controversy about the prognostic nature of PD-L1 expression and TILs and their clinicopathological relevance in patients with TNBC (13-20).
Therefore, there is an urgent need to determine the prognostic values of tumoral PD-L1, TILs, and their associations with the clinicopathological features in TNBC patients. To the best of our knowledge, this meta-analysis for the first time has aimed to determine the prognostic values of tumoral PD-L1, TILs, and their associations with clinicopathological features in TNBC patients.
Discussion
Multiple studies have investigated the association between tumoral PD-L1 expression and TILs with the prognosis of patients with TNBC. However, there is a controversy about the prognostic nature of PD-L1 expression and TILs in patients with TNBC (13-18). As far as we know, it is the first meta-analysis that has intended to resolve this controversy.
Our analysis has displayed that there has been no association between tumoral PD-L1 expression and better prognosis in patients with TNBC. Although a meta-analysis has shown that tumoral PD-L1 expression is associated with poor prognosis in patients with breast cancer (HR, 1.63; 95% CI, 1.07–2.46; P = 0.02), this study has failed to demonstrate any significant HR between tumoral PD-L1 expression and better prognosis in TNBC patients. (P-value ≥ 0.05) (28). Since the tumor microenvironment of TNBC is regulated with multiple immune checkpoints and immune cells, the measurement of only PD-L1 expression might not be a solid prognostic value (10). Indeed, PD-L1, on its own, may not serve as a reliable prognostic biomarker. However, it has a notable prediction value when reported with other indicators, e.g., TILs (29).
Our study has demonstrated that a high level of TILs is associated with increased long-term OS and DFS in TNBC patients. Consistent with our study, another meta-analysis has highlighted that the high level of TILs is associated with a better prognosis in TNBC patients (19). Moreover, Gao et al. has reported that increased TILs conferred improved survival rate. In patients with TNBC, CD4+, CD8+, and FOXP3+ lymphocytes have been indicative of a favorable prognosis (30).
We have also analyzed the potential association between PD-L1 expression and TILs with clinicopathological features in TNBC patients. According to our pooled analysis, there has been no significant association between PD-L1 expression and clinicopathological features (all P-values were higher than 0.05 (P-value ≥ 0.05)). Furthermore, there has been no significant association between TILs and clinicopathological features. Therefore, increased TILs have not significantly pertained to increased tumor size, higher tumor grade, and positive lymph node metastasis.
Moreover, in the current study, we have shown a strong association between PD-L1 expression and TILs. Consistent with this, Byung Bae et al. have discovered that high expression of PD-L1 was significantly related to high TILs levels (31). Increased TILs levels eliminate the tumoral cells and can predict a better response rate to immunotherapy. Increased PD-L1 expression and up-regulated level of TILs have provided ample opportunity for the immunotherapy to eliminate the tumoral cells (32). Consistent with this, immune checkpoint therapy has shown promising results in patients with TNBC (33).
This current meta-analysis has several strengths. First, as far as we can tell, it is the first meta-analysis that has comprehensively elucidated the impact of TILs and tumoral PD-L1 expression on prognosis and clinicopathological features of TNBC patients. Second, this study has highlighted a strong association between tumoral PD-L1 expression and TILs. Third, highlighting this strong association for cancers like TNBC, which they do not have well-established targets, is even more valuable. However, this meta-analysis has several limitations. First, only the papers in English were included, while other published records were omitted. Second, a small number of studies were included in the analysis, i.e., studies evaluated TILs with different methods (IHC and H&E), and the studies utilized variable cut-off values for assessing PD-L1 expression, and investigations used different scoring systems for evaluating TILs. These mentioned limitations probably lead to heterogeneity between the included studies.
Conclusion
This study has intended to highlight the prognostic values of TILs and tumoral PD-L1 expression and their associations with clinicopathological features in patients with TNBC. According to this study, increased level of TILs has been associated with better prognosis in TNBC patients. From a clinical perspective, there has been a strong association between the tumoral PD-L1 expression and TILs. This strong association can serve as a pivotal target for cancer immunotherapy in patients with TNBC.
Round 2
Reviewer 2 Report
The authors have adequately addressed most Reviewers’ comments and the revised manuscript is significantly improved. The authors itemized briefly their responses to the various remarks of the Reviewer, as requested. Overall, the current version of the paper is well-written and contributes to the existing knowledge in its field. In conclusion, the Reviewer is convinced that this revised manuscript is now suitable and competitive enough for publication in “Diagnostics”.